# Adaptive Posture-Balance Cardiac Rehabilitation Exercise Significantly Improved Physical Tolerance in Patients with Cardiovascular Diseases

**DOI:** 10.3390/jcm11185345

**Published:** 2022-09-12

**Authors:** Mei Ma, Bowen Zhang, Xinxin Yan, Xiang Ji, Deyu Qin, Chaodong Pu, Jingxiang Zhao, Qian Zhang, Heinz Lowis, Ting Li

**Affiliations:** 1Institute of Biomedical Engineering, Chinese Academy of Medical Sciences and Peking Union Medical College, Tianjin 300192, China; 2Department of Rehabilitation Medicine, Tianjin Chest Hospital, Tianjin 300192, China; 3Department of Cardiology, Key Laboratory of Pulmonary Vascular Medicine, Fuwai Hospital, Chinese Academy of Medical Sciences and Peking Union Medical College, Beijing 100037, China; 4Drei-Burgen-Klinik of German Pension Insurance of Rhineland-Palatinate, 55583 Bad Kreuznach, Germany

**Keywords:** cardiac rehabilitation, exercise therapy, balance exercises, cardiovascular diseases

## Abstract

Cardiac rehabilitation (CR) requires more professional exercise modalities to improve the efficiency of treatment. Adaptive posture-balance cardiac rehabilitation exercise (APBCRE) is an emerging, balance-based therapy from clinical experience, but lacks evidence of validity. Our study aimed to observe and assess the rehabilitation effect of APBCRE on patients with cardiovascular diseases (CVDs). All participants received one-month APBCRE therapy evenly three times per week and two assessments before and after APBCRE. Each assessment included cardiopulmonary exercise testing (CPET), resting metabolic rate (RMR) detection, and three questionnaires about general health. The differences between two assessments were analyzed to evaluate the therapeutic effects of APBCRE. A total of 93 participants (80.65% male, 53.03 ± 12.02 years) were included in the analysis. After one-month APBCRE, oxygen uptake (VO_2_, 11.16 ± 2.91 to 12.85 ± 3.17 mL/min/kg, *p* < 0.01) at anaerobic threshold (AT), ventilation (VE, 28.87 ± 7.26 to 32.42 ± 8.50 mL/min/kg, *p* < 0.01) at AT, respiratory exchange ratio (RER, 0.93 ± 0.06 to 0.95 ± 0.05, *p* < 0.01) at AT and oxygen uptake efficiency slope (OUES, 1426.75 ± 346.30 to 1547.19 ± 403.49, *p* < 0.01) significantly improved in CVD patients. The ≤55-year group had more positive improvements (VO_2_ at AT, 23% vs. 16%; OUES, 13% vs. 6%) compared with the >55-year group. Quality of life was also increased after APBCRE (47.78 ± 16.74 to 59.27 ± 17.77, *p* < 0.001). This study proved that APBCRE was a potentially available exercise rehabilitation modality for patients with CVDs, which performed significant increases in physical tolerance and quality of life, especially for ≤55-year patients.

## 1. Introduction

Cardiovascular diseases (CVDs) are a group of disorders of heart and blood vessel disorders, such as coronary heart disease [1] and heart failure [2]. CVDs are the leading cause of death and disability in the world [3]. Approximately 330 million people in China suffer from CVDs, and the prevalence continues to increase [4]. With the progression of CVDs, physical tolerance in patients gradually declines until complete loss, along with increasing dyspnea. It seriously affects the quality of life of patients and causes a huge social burden [5]. Therefore, it is critical to improve the physical tolerance of CVDs patients.

The American College of Cardiology guidelines emphasize [6] that cardiac rehabilitation (CR) is an important and effective approach to preventing and treating CVDs and is strongly recommended for clinical practice. Moreover, numerous studies confirm [7,8] that CR improves physical tolerance [9] and quality of life in patients with CVDs [10,11], reduces the incidence risk of CVDs [12], and decreases the rate of hospital readmission.

At present, the main modality of CR is physical exercise [13], occasionally supplemented with health education and psychological counseling [7]. Patients could choose to finish CR at home or in center depending on their condition. Although there are no significant differences in rehabilitation effects between home-based CR and center-based CR, home-based CR provides better satisfaction and comfort for patients [14,15]. The general exercise methods of CR [16] include walking, jogging, cycling and other aerobic exercises, combined with resistance training. Emerging techniques and traditional exercises have been explored and are found to be effective, such as high-intensity interval training (HIIT) [17,18], yoga [19,20] and Tai Chi [21]. In most studies, the exercise intensity is controlled at 40% to 80% of the maximum heart rate (HR) [22,23], and the exercise frequency ranges from 3 to 6 times per week [24,25]. However, the above sports methods and rehabilitation modalities are mainly based on existing general exercise models [6,17], rather than exclusively focusing on CVDs. Universal models usually do not achieve the expected effect due to lack of pertinence, although they are intensively adopted.

Based on balance exercise, a new rehabilitation approach was designed and named as adaptive posture-balance cardiac rehabilitation exercise (APBCRE). The approach was inspired by clinical practice about CVDs, specifically designed to reduce the risk of falls. To better understand the clinical effectiveness of APBCRE on CVDs patients, the current study aimed to assess whether APBCRE could enhance physical tolerance and improve the quality of life in patients with CVDs.

## 2. Materials and Methods

### 2.1. Study Design

This experiment was performed among CVDs patients from December 2020 to March 2021 in Tianjin Chest Hospital. This study protocol was approved by the local ethics committee (IRB-SOP-016(F)-001-02, 9 August 2021). All subjects have signed informed consent forms before being enrolled. The whole experiment included one month of APBCRE and two clinical assessments before and after APBCRE interventions. The one-month APBCRE consisted of twelve exercise sessions, evenly three times per week. Each assessment included cardiopulmonary exercise testing (CPET), resting metabolism rate (RMR) detection, and questionnaires about quality of life (QoL), depression levels and anxiety levels. The primary outcome was physical tolerance assessed by oxygen uptake (VO_2_) at anaerobic threshold (AT). The secondary endpoints were the resting metabolism level and QoL measured by RMR and 12-item Short Form Survey (SF-12), respectively. A flowchart of this study is provided in Figure 1.

### 2.2. Patient Selection

Subjects were recruited from patients with CVDs. The inclusion criteria were (1) age over 18 years old; (2) diagnosed as CVDs, including coronary heart disease (CHD), old myocardial infarction (MI), arrhythmias and heart valve disease; (3) without percutaneous coronary intervention (PCI) or one week after PCI; (4) without coronary artery bypass graft (CABG) or one month after CABG. Patients were excluded if having abnormal blood pressure response, acute heart failure, unstable angina, acute myocardial infarction, congenital heart disease, and severe musculoskeletal diseases limiting [23].

### 2.3. Rehabilitation Protocol

Depending on personal physical conditions, participants were assigned to three danger levels: low-level, medium-level, and high-level. The standards of danger level are shown in Table A1 in Appendix A. The different danger-level patients underwent individualized APBCRE with matched different exercise thresholds and accepted comprehensive guidance from professional nurses.

The fundamental process of APBCRE consisted of four parts: breathing training and warm-up, aerobic exercise, resistance exercise, and flexibility exercise. The first part generally lasted 5–15 min for any danger level. For better effects of sport rehabilitation, our study designed a new warm-up method based on balance exercise, which was the essence of APBCRE. Figure 2a outlined the specific steps of the new warm-up method, including stretching of upper limbs, legs, waist, and other parts. The first part mainly contributed to improving body coordination and balance. The second part was moderate-intensity endurance exercise. The intensity was controlled at 40–60% AT, 60–70% of peak HR and Borg grade 12–13. The exercise duration of aerobic exercise was usually 30 min. Moreover, a body-building vehicle was used for resistance exercise in the third part for 10–15 min. The resistance power of the bicycle was adjusted depending on danger level and VO_2_ at AT. The last part was the continuation of low-intensity aerobic training for 5–10 min. It was designed to slow flow of blood from the skeletal muscles back to the heart, which could effectively prevent a significant increase in cardiac stress. In summary, the total exercise time for one session was generally 50–70 min, varying with physical function of patients. Although there is no specific date for each training, patients were required to complete 12 sessions within one month.

### 2.4. Outcome Measure

Cardiopulmonary Exercise Testing (CPET)

CPET was performed on the exercise cardiopulmonary function measurement system (Oxycon Mobile, JAEGER-CareFusion, Hoechberg, Germany) (CPX, Figure 2b). Individualized ramp protocol was used for CPET. HR, VO_2_, respiratory exchange ratio (RER) and ventilation (VE) were collected at resting state and AT, respectively. AT was defined by the V-slope method. VE-VCO_2_ slope (VE/VCO_2_) and oxygen uptake efficiency slope (OUES) was calculated based on VO_2_, VE and carbon dioxide output (VCO_2_). The power of the bicycle at AT (WAT) in CPET was also collected to evaluate sports performance in participants. Maximum effort was reached when RER was above 1.05.

Resting Metabolic Rate (RMR)

Resting metabolic rate was also measured by CPX. The energy expenditure (ee) of RMR was calculated by detecting VO_2_ and VCO_2_ in the resting state. The equation was ‘ee = 1.59 × VCO_2_ + 5.68 × VO_2_ + 2.17 × α^2^’, in which α was a fixed variable depending on patients. RMR consisted of three parts: fat energy (fat), carbohydrate energy (cho) and protein energy. Protein energy was set as a constant (405 Kcal/d), and the others were computed as ee.

General health assessment of quality of life, anxiety and depression

Three validated questionnaires were used to assess general health of participants. It included quality of life assessed by SF-12, level of anxiety assessed by Generalized Anxiety Disorder 7-item (GAD-7), level of depression assessed by Patient Health Questionnaire-9 (PHQ-9). The score of SF-12 was a continuous variable, ranging from 0 to 100. Closer to 0 meant lower quality of life, and closer to 100 was opposite. GAD-7 and PHQ-9 were grade variables, which were respectively divided into 4 groups and 5 groups. Higher score presented higher severity of anxiety or depression.

### 2.5. Sample Size

Sample size calculation was performed for primary outcome physical tolerance measured by VO_2_ at AT. Michitaka K. et al. [16] found that VO_2_ at AT notably increased around 11.7% (11.1 ± 1.1 to 12.4 ± 2.4 mL/min/kg) after rehabilitation. We hypothesized that significance level was 0.05, power was 0.90 and the improvement of before and after intervention was 15%. The sample size was calculated as at least 21 participants per group. Since our study was a self-controlled experiment, at least 21 patients were needed in total. The sample size was calculated using online free tool from Harvard University (http://hedwig.mgh.harvard.edu/sample_size/js/js_parallel_quant.html, accessed on 7 October 2020).

### 2.6. Statistical Analysis

Continuous variables were described by mean and standard deviation (SD), and categorical variables were described by absolute count and relative frequency. K-NearestNeighbor (KNN) algorithm was used to fill in the missing values. The differences of continuous variables between before and after APBCRE were compared by two-tailed paired Student’s *t* test. Fisher’s exact test was used for categorical variables. Moreover, all participants were divided into two subgroups (≤55-year group and >55-year group) by the mean age in order to analyze age differences in rehabilitation effect of APBCRE. The same analysis methods were applied to compare differences within subgroups between two time points. We also counted the rate of changes in outcomes for each patient after APBCRE, which aimed to compare the alteration degree of multiple indicators.

Two-tailed *p* < 0.05 was regarded as the significant level for all tests. Data analyses and visualization were conducted with R (version 3.6.2, created by Robert Clifford Gentleman and George Ross Ihaka, https://www.r-project.org/, accessed on 30 July 2022) and Python (version 3.7, created by Guido van Rossum, https://www.python.org/, accessed on 30 July 2022). 

## 3. Results

### 3.1. Participants Characteristics

In our study, 93 enrolled patients were all eligible for analysis. Table 1 outlines the demographic characteristics and clinical profiles of patients. Overall, 80.65% of patients were male and the mean age was 53.03. Most of the participants (77.42%) were overweight (body mass index (BMI) > 24.0 kg/m^2^), even 29.03% obesity (BMI ≥ 28.0 kg/m^2^). In CVDs composition, 72 (77.42%) patients had CHD, 47 (50.54%) had MI, 21 (22.58%) had arrhythmias and 7 (7.53%) had heart valve disease. Of these, 44 patients were complicated with hypertension, and 17 with diabetes. More than one-third of patients (44.09%) have accepted PCI, and 14 patients have undergone CABG.

### 3.2. Physical Tolerance

VO_2_ at AT increased significantly after one-month APBCRE (11.16 ± 2.91 to 12.85 ± 3.17 mL/min/kg, *p* < 0.01) (Table 2). VE at AT, RER at AT were also significantly different (respectively, 28.87 ± 7.26 to 32.42 ± 8.50 mL/min/kg, *p* < 0.001; 0.93 ± 0.06 to 0.95 ± 0.05, *p* < 0.01). Moreover, the variation of VE was higher than VO_2_ (3.55 vs. 1.69 mL/min/kg). VE at AT and VO_2_ at AT had higher changing proportions (more than 15%) compared with other notably different variables. There were no significant differences between before and after APBCRE in resting state (all *p* > 0.05).

To explore the specific efficiency of APBCRE in different age groups, participants were divided into ≤55-year group and >55-year group by the average age. The ≤55-year group contained 54 patients (49 male, 44.67 ± 6.73 years), and the >55-year group contained 39 patients (26 male, 64.62 ± 7.04 years). VO_2_ at AT increased significantly in both groups (*p* < 0.01) (Figure 3a), while the ≤55-year group had higher changing proportion (0.23 (95%CI, 0.1 to 0.35)) compared with >55-year group (0.16 (95%CI, 0.09 to 0.23)) (Figure 4). But the rate of change of VE at AT was similar in two subgroups (0.17 (95%CI, 0.06 to 0.28) vs. 0.17 (95%CI, 0.08 to 0.26)). OUES was significantly different (1531.19 ± 265.11 to 1706.60 ± 363.39, *p* < 0.01) in the ≤55-year group, but not different in the >55-year group (*p* = 0.22). More details about CPET results being significantly different were shown in Figure 3, including VE at AT, VO_2_ at AT, RER at AT and OUES.

### 3.3. Secondary Endpoints

The resting metabolic rate was not significantly different between before and after APBCRE (Figure 5a), including total energy, fat energy, and carbohydrate energy. However, the score of SF-12 significantly increased after one-month APBCRE (47.78 ± 16.74 to 59.27 ± 17.77, *p* < 0.001) (Figure 5b). The level distribution of PHQ-9 also varied significantly (*p* < 0.05), but the GAD-7 had no difference (*p* = 0.06, data not shown). The number of PHQ-9 scores below 10 changed from 26 (83.87%) to 29 (93.55%). WAT also increased significantly after APBCRE intervention (56.56 ± 23.55 to 68.85 ± 24.46 watt, *p* < 0.001) (Figure 5c).

## 4. Discussion

Our study demonstrated that APBCRE was a potentially safe and effective rehabilitation approach for patients with CVDs. Patients performed a significant increase in physical tolerance after undergoing one-month APBCRE. The ≤55-year group was more positive than the >55-year group. Quality of life and level of anxiety were also notably improved. APBCRE is the combination of existing exercise modalities and traditional medicine. It starts from respiratory regulation, and gradually extends the limb movement to the whole body through aerobic exercise, resistance exercise and flexibility training. APBCRE aims to improve neuroplasticity of the autonomic nerve through resetting the pattern of exercise.

European and American Heart Disease guidelines [26,27] recommend exercise rehabilitation as an adjuvant treatment for CVDs to compensate some shortcomings of pharmacological therapy. It is universally accepted that exercise rehabilitation is beneficial to improving physics tolerance [6], although there is controversial in specific exercise modalities and intensity [23]. In previous studies, physics tolerance is usually assessed by the peak oxygen uptake (VO_2peak_) [8,10,28]. However, VO_2peak_ needs to be measured in the exhaustion state, which is easily interfered by subjective consciousness. Therefore, our study chose VO_2_ at AT instead of VO_2peak_ to ensure the objectivity of measurements. We found that VO_2_ at AT significantly improved by 19.79%, which was similar to other rehabilitation modalities (simple aerobic exercise [29] and HIIT [17]). Meanwhile, it confirmed the positive rehabilitation effect of APBCRE.

Moreover, we observed that VO_2_, VE at AT in two age subgroups both significantly increased, while the ≤55-year group improved more. OUES only increased in the ≤55-year group (*p* < 0.01 vs. *p* = 0.22). OUES was an objective, reproducible measure of cardiopulmonary reserve, which integrated cardiovascular, musculoskeletal and respiratory function [30]. The differences between subgroups indicated that APBCRE had various modes of effect for different age levels. For lower-age patients, APBCRE improved both musculoskeletal, respiratory and cardiovascular function. However, for higher-age patients, APBCRE mainly enhanced ventilation when sporting rather than directly improving oxygen utilization of skeletal muscle. The improvement of ventilation was also relatively constrained. This result was consistent with the irreversible alterations in skeletal muscles and myocardium from aging. Thus, age is a nonnegligible factor when making exercise rehabilitation protocol for CVDs patients.

Furthermore, our study showed the positive therapeutic effect of exercise rehabilitation on elder patients with CVDs, which was similar to the results of Marchionni et al. [31] and Campo G et al. [32]. Lachman S et al. [33] found that moderate exercise training contributes to improving cardiovascular functions, even for elderly patients. These results confirmed that appropriate physical exercise played an important role in preventing and treating CVDs without age limitation.

The other important purpose of CR is to improve the quality of life [31], which is directly perceived by patients. Our study made individualized APBCRE programs and professional guidance for each participant to ensure more suitable exercise intensity and sports modality. The results showed that one-month APBCRE effectively improved quality of life, depression level and sports performance. However, the finding in previous studies was controversial. Snoek J.A. et al. [3] showed no differences in quality of life between the home-based mobile-guided cardiac rehabilitation group and controlled group. Yan-Wen Chen et al. [10] observes the opposite result in patients with chronic heart failure. It is indicated that the paradox possibly results from different types of CVDs and diverse sports modalities. Therefore, we will conduct additional experiments to verify the effectiveness of APBCRE on the quality of life of CVDs patients in the future.

In addition, cardiac function indicators or metabolic rate in resting state had no notable alterations after one-month APBCRE. The differences between AT and resting state indicated that short-term exercise rehabilitation mainly improved compensation capacity when sporting and had limited benefit for the whole organic function and basal metabolism. Eva Prescott et al. [5] showed that the rehabilitation efficacy was not well maintained at one year compared with the end of exercise. Therefore, we suggested that long-term regular rehabilitation was essential to improving overall function of the cardiovascular system.

There are some limitations in this study. Firstly, all patients in our study were recruited from a single center, and the sample distributions of gender and age were unbalanced. It limited the observation of the outcome of female and elderly patients, especially those over 75 years of age. Secondly, our study was a self-controlled experiment without the non-intervention control group. It led to a moderate decrease in the precision and explanation of experiments. Finally, the advantages of APBCRE were not fully explored due to a lack of comparing APBCRE with other exercise modalities. In the future, we plan to conduct a multi-center randomized controlled trial with more samples to cover the shortcomings of this study and further confirm our findings.

## 5. Conclusions

This study showed that the self-created rehabilitation method (adaptive posture-balance cardiac rehabilitation exercise, APBCRE) significantly improved the physical tolerance and quality of life of patients with CVDs. Moreover, compared with the >55-year group, the ≤55-year had more positive therapeutic efficiency.

## Figures and Tables

**Figure 1 jcm-11-05345-f001:**
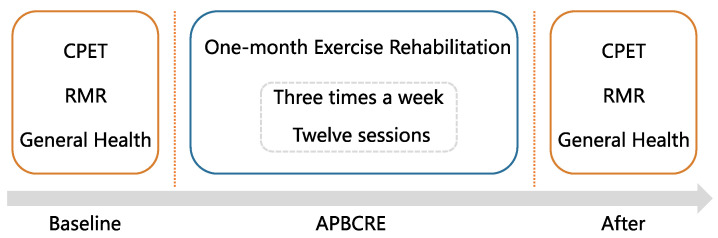
Flowchart of this study. CPET cardiopulmonary exercise testing, RMR resting metabolism rate, APBCRE adaptive posture-balance cardiac rehabilitation exercise.

**Figure 2 jcm-11-05345-f002:**
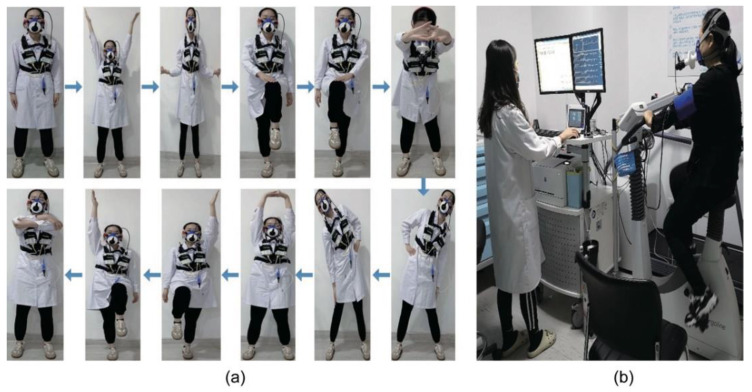
(**a**) Critical steps of balance exercise in APBCRE; (**b**) operation diagram of CPX. APBCRE adaptive posture-balance cardiac rehabilitation exercise. CPX exercise cardiopulmonary function measurement system.

**Figure 3 jcm-11-05345-f003:**
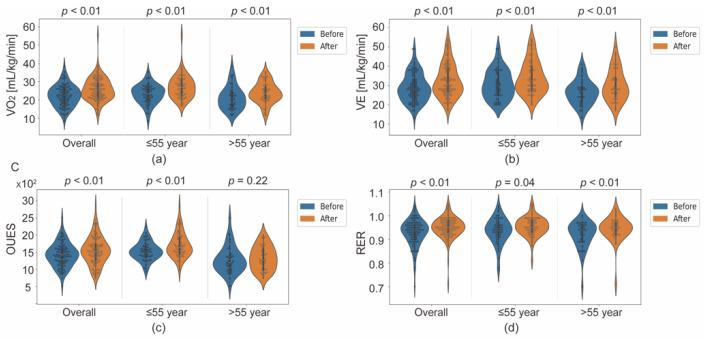
Distribution of significantly different variables in CPET between before and after APBCRE in all participants, the ≤55-year group and the >55-year group. (**a**) VO_2_ at AT; (**b**) VE at AT; (**c**) OUES; (**d**) RER at AT. CPET cardiopulmonary exercise testing, APBCRE adaptive posture-balance cardiac rehabilitation exercise, VO_2_ oxygen uptake, VE ventilation, OUES: oxygen uptake efficiency slope, RER respiratory exchange ratio, AT anaerobic threshold.

**Figure 4 jcm-11-05345-f004:**
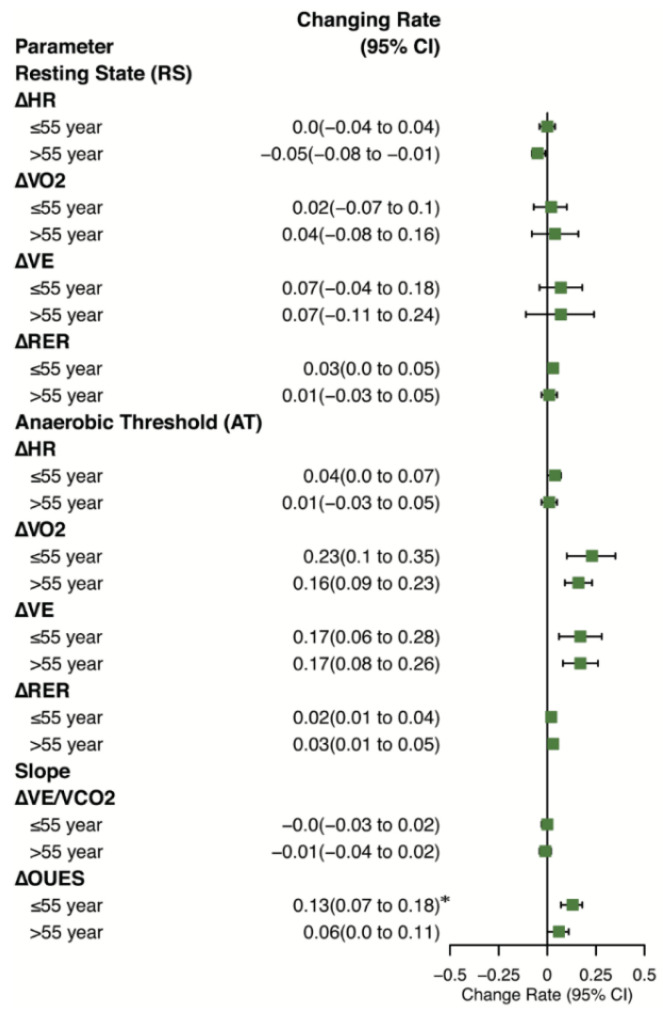
Changing rate of the ≤55-year group and >55-year group about CPET. * *p* < 0.05, comparing the increment between the ≤55-year group and >55-year group. Δ refers to the changing rate of variables. CPET cardiopulmonary exercise, HR heart rates, VO_2_ oxygen uptake, RER respiratory exchange ratio, VE ventilation, VE/VCO_2_ VE–VCO_2_ slope, OUES, oxygen uptake efficiency slope, CI Confidence interval.

**Figure 5 jcm-11-05345-f005:**
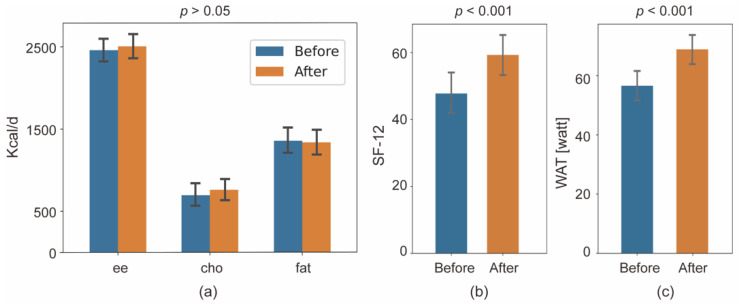
Secondary endpoint results before and after APBCRE: (**a**) resting metabolic rate, including energy expenditure, carbohydrate energy and fat energy; (**b**) the score of SF-12; (**c**) bicycle power at AT. APBCRE adaptive posture-balance cardiac rehabilitation exercise, AT anaerobic threshold.

**Table 1 jcm-11-05345-t001:** Characteristics of study participants.

Characteristics	Normal
Sex (Male)	75 (80.65%)
Mean age (years)	53.03 (12.02)
Mean body mass index (kg/m^2^)	
<18.5	3 (3.22%)
18.5~24.0	18 (19.35%)
24.0~28.0	45 (48.39%)
≥28.0	27 (29.03%)
Coronary Heart Disease (%)	72 (77.42%)
Old Myocardial Infarction (%)	47 (50.54%)
Arrhythmias (%)	21 (22.58%)
Heart Valve disease (%)	7 (7.53%)
Hypertension (%)	44 (47.31%)
Diabetes (%)	17 (18.28%)
Percutaneous Coronary Intervention (%)	41 (44.09%)
Coronary Artery Bypass Graft (%)	14 (15.05%)

**Table 2 jcm-11-05345-t002:** CPET parameters at before and after rehabilitation in different groups.

	Overall					≤55 year					>55 year				
	Before		After		Before vs. After *p* (*t* Test)	Before		After		Before vs. After *p* (*t* Test)	Before		After		Before vs. After *p* (*t* Test)
	Mean	SD	Mean	SD		Mean	SD	Mean	SD		Mean	SD	Mean	SD	
Resting State (RS) ^a^															
HR (cpm)	78.52	11.88	76.18	10.70	0.06	78.54	11.98	77.83	11.51	0.65	78.49	11.88	73.90	9.13	0.01 *
VO_2_ (mL/min/kg)	4.33	1.18	4.25	1.28	0.57	4.20	0.95	4.12	1.10	0.63	4.52	1.43	4.42	1.50	0.73
VE (mL/min/kg)	13.22	4.44	13.11	4.53	0.84	13.43	4.71	13.50	4.86	0.91	12.92	4.06	12.56	4.02	0.68
RER	0.81	0.07	0.82	0.08	0.11	0.80	0.06	0.82	0.07	0.07	0.82	0.07	0.83	0.09	0.63
Anaerobic Threshold (AT) ^b^															
HR (cpm)	104.03	15.55	105.81	14.04	0.21	105.19	14.76	108.19	14.38	0.10	102.44	16.65	102.51	13.03	0.97
VO_2_ (mL/min/kg)	11.16	2.91	12.85	3.17	0.00 **	11.58	2.64	13.54	3.25	0.00 **	10.58	3.19	11.90	2.81	0.00 **
VE (mL/min/kg)	28.87	7.26	32.42	8.50	0.00 **	30.59	7.46	34.02	8.51	0.01 *	26.49	6.33	30.21	8.07	0.00 **
RER	0.93	0.06	0.95	0.05	0.00 **	0.94	0.05	0.95	0.04	0.02 *	0.92	0.06	0.94	0.05	0.00 **
Slope															
VE/VCO_2_	29.94	5.18	29.69	5.42	0.58	28.36	3.33	28.17	3.69	0.61	32.12	6.40	31.80	6.65	0.56
OUES	1426.75	346.30	1547.19	403.49	0.00 **	1531.19	265.11	1706.60	363.39	0.00 **	1282.14	394.15	1326.47	351.94	0.22

* *p* < 0.05; ** *p* < 0.01; when comparing. ^a^ at the beginning of the whole test with rest state. ^b^ in the process of the whole test reaching the critical value of AT. SD standard deviation, cpm counts per minutes, HR heart rates, VO_2_ oxygen uptake, RER respiratory exchange ratio, VE ventilation, VE/VCO_2_ VE–VCO_2_ slope, OUES oxygen uptake efficiency slope.

## Data Availability

The data used in this study are available from the corresponding author upon reasonable request.

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
