# Peer review of "Adaptive Posture-Balance Cardiac Rehabilitation Exercise Significantly Improved Physical Tolerance in Patients with Cardiovascular Diseases"

_jcm, 2022, doi:10.3390/jcm11185345_

Round 1

Reviewer 1 Report

This is an interesting study about the use of Adaptive posture-balance cardiac rehabilitation exercise to improve exercise capacity in patients with CVD. I have some questions:

In abstract: It is not clear the significance of this sentence “The factor of 24 age was addressed in data analysis as well”. Being this an abstract I suggest to remove it (results are already explicative).

Line 40, page 1 introduction: please remove “etc”.

English revision is needed.

In introduction use present simple. 

Please quote PMID: 31806139 as another example of tailored exercise to improve functional capacity in older adults with previous CV disease. 

Please define in methods not only exclusion criterion, but also the clinical characteristics of patients with CVD to be included (e.g were patients enrolled after how much time from an acute coronary syndrome?) 

How was assigned the level of danger?

Being this a prospective study a sample size calculation should be present in methods.

Please remove this sentence “Authors should discuss the results and how they can be interpreted from the perspective of previous studies and of the working hypotheses. The findings and their implications should be discussed in the broadest context possible. Future research directions may also be highlighted” from the beginning of the discussion.

Author Response

Response to Reviewer 1 Comments

Manuscript ID number: jcm-1866806

Title: Adaptive posture-balance cardiac rehabilitation exercise significantly improved physical tolerance in patients with cardiovascular diseases

Authors: Mei Ma, Bowen Zhang, Xinxin Yan, Xiang Ji, Deyu Qin, Chaodong Pu, Jingxiang Zhao, Qian Zhang, Heinz Lowis, Ting Li *

We are grateful to your professional review work on our article. As you are concerned, there are several problems that need to be addressed. According to your nice suggestions, we have made extensive corrections to our previous draft. The detailed corrections are listed below.

Because almost every sentence was revised in English, it is messy to highlight all revisions. Thus, We highlight all revisions in content by yellow color, and some important revisions in English by blue color. Thank you very much for understanding.

Point 1: In abstract: It is not clear the significance of this sentence “The factor of 24 age was addressed in data analysis as well”. Being this an abstract I suggest to remove it (results are already explicative).

Response 1: Thank you for the suggestion. We apologize for this vague description. We have removed the sentence “The factor of age was addressed in data analysis as well” in abstract.

Point 2: Line 40, page 1 introduction: please remove “etc”.

Response 2: Thank you for the suggestion. We have removed “etc” in line 40.

Point 3: English revision is needed.

Response 3: We are very apologetic about our poor writing. We tried our best to check every sentence in our draft. We take great care to revise the grammar, typos and professional representation of our manuscript. And we invited a native English speaker to polish the manuscript. Due to many changes, we cannot list them all here. But we have marked them in blue in the revised paper. Thank you for understanding. And we hope the revised manuscript could be more clear to you.

Some changes are listed here as examples:

  1. Our group in Tianjin Chest Hospital designed a peculiar rehabilitation method based on balance exercise, named as adaptive posture-balance cardiac rehabilitation exercise (APBCRE). It originated from the summary of large number of daily clinical work experience about CVDs and was specifically designed for reducing the risk of falls. Nevertheless, the actual application effect of APBCRE on patients was not definite. Thus, this study aimed to assess whether this self-created therapy is effective on patients with cardiovascular diseases, and observe concrete changes in physiology and behavior.” is changed toBased on balance exercise, a new rehabilitation approach was designed and named as adaptive posture-balance cardiac rehabilitation exercise (APBCRE). The approach was inspired by clinical practice about CVDs, specifically designing to reduce the risk of falls. To better understand the clinical effectivness of APBCRE on CVDs patients, the current study aimed to assess whether APBCRE could enhance physical tolerance and improve the quality of life in patients with CVDs.in introduction in lines 63-68.
  2. Before being enrolled, all subjects have signed an informed consent form. The whole experiment included one-month APBCRE therapy and two assessments. The one-month APBCRE consisted of twelve exercise sessions, evenly three times per week. At baseline and after rehabilitation, participants underwent one assessment, including cardiopulmonary exercise testing (CPET), resting metabolic rate (RMR) detection, and quality of life questionnaire (QoL).” is changed to “All subjects have signed informed consent forms before being enrolled. The whole experiment included one-month APBCRE and two clinical assessments before and after APBCRE interventions. The one-month APBCRE consisted of twelve exercise sessions, evenly three times per week. Each assessment included cardiopulmonary exercise testing (CPET), resting metabolism rate (RMR) detection, and questionnaires about quality of life, depression levels and anxiety levels.“ in method in lines 73-78.
  3. Therefore, the different results of two subgroups indicated that this rehabilitation approach had diverse effect modes on different age levels. For lower-age patients, exercise rehabilitation improved musculoskeletal, respiratory function and cardiovascular, while for higher-age patients, the improvement mainly focused on respiratory function and the increment was also relatively limited.” is changed to “The differences between subgroups indicated that APBCRE had various modes of effect for different age levels. For lower-age patients, APBCRE improved both musculoskeletal, respiratory and cardiovascular function. But for higher-age patients, APBCRE mainly enhanced ventilation when sporting rather than directly improving oxygen utilization of skeletal muscle.” in discussion in lines 247-251.

Point 4: In introduction use present simple.

Response 4: Thank you for the suggestion. We feel very sorry for our poor writing about tense. We have changed introduction to present simple, except for contents about our study. We think these contents are more appropriate for the general past tense as the description of our finished work.

The main revisions are below:

  1. CVDs have become the primary reason for death and disability in the whole world. In China, about 3.3 billion persons were suffering from CVDs and the prevalence was still continuously increasing.” was changed to “CVDs are the leading cause of death and disability in the world. Approximately 3.3 billion people in China suffer from CVDs, and the prevalence continues to increase” in lines 38-40.
  2. The American College of Cardiology guidelines emphasized that cardiac rehabilitation (CR) was an important preventive mean and effective therapeutic approach for CVDs, and strongly recommended for clinical use.” is changed to “The American College of Cardiology guidelines emphasize that cardiac rehabilitation (CR) is an important and effective approach to preventing and treating CVDs and is strongly recommended for clinical practice.” in lines 44-46.
  3. Many studies have also shown that CR could improve exercise tolerance and quality of life of patients, reduce the incidence risk of CVDs, and decrease the rate of hospital readmission.” is changed to “Moreover, numerous studies confirm that CR improves physical tolerance and quality of life in patients with CVDs, reduces the incidence risk of CVDs, and decreases the rate of hospital readmission.” in lines 46-48.
  4. At present, the main modality of CR was physical exercise, occasionally complemented with health education and psychological counseling” is changed to “At present, the main modality of CR is physical exercise, occasionally supplemented with health education and psychological counseling” in lines 49-50.
  5. There was no significant differences between two types in rehabilitation effect, while home-based CR provided better satisfaction and comfort for patients” is changed to “Although there are no significant differences in rehabilitation effects between home-based CR and center-based CR, home-based CR provides better satisfaction and comfort for patients” in lines 51-53.
  6. The common exercise methods included walking, jogging, cycling, and other aerobic exercises, sometimes combined with resistance training. Some new methods and Chinese traditional exercise have also been gradually explored and showed significant effects, such as high-intensity interval training (HIIT), yoga, Tai Chi.” is changed to “The general exercise methods of CR include walking, jogging, cycling and other aerobic exercises, as well as combined with resistance training. Emerging techniques and traditional exercises have been explored and found to be effective, such as high-intensity interval training (HIIT), yoga and Tai Chi” in lines 53-57.
  7. In most studies, the exercise intensity was usually controlled at 40% to 80% of the maximum heart rate, and the exercise frequency was from 3 times per week to 6 times per week.” is changed to “In most studies, the exercise intensity is controlled at 40% to 80% of the maximum heart rate, and the exercise frequency ranges from 3 times per week to 6 times per week.” in lines 57-59.
  8. However, the above rehabilitation modes and training modalities were generally based on existed common exercise methods, not focusing on specific diseases.” is changed to “However, the above sports methods and rehabilitation modalities are mainly based on existing general exercise models, rather than exclusively focusing on CVDs.” In lines 59-61.
  9. Although traditional methods were intensively adopted, due to lack of pertinence, the rehabilitation usually could not achieve the expected effect.” is changed to “Universal models usually do not achieve the expected effect due to lack of pertinence, although they are intensively adopted.” In lines 61-62.

Point 5: Please quote PMID: 31806139 as another example of tailored exercise to improve functional capacity in older adults with previous CV disease.

Response 5: We are very grateful to this suggestion. We have added this article in dicussion in line 257 “which was similar to the results of Marchionni et al [32] and Campo G et al [32]” as another support for the result “positive therapeutic effect of exercise rehabilitation on elder patients with CVDs “. In addition, we have checked the literature carefully and cited another study “Campo, G.; Tonet, E.; Chiaranda, G.; Sella, G.; Maietti, E.; Bugani, G.; Vitali, F.; Serenelli, M.; Mazzoni, G.; Ruggiero, R.; et al. Exercise Intervention Improves Quality of Life in Older Adults after Myocardial Infarction: Randomised Clinical Trial. Heart 2020, 106, 1658–1664.” in introduction in line 47 to enrich background description.

Point 6: Please define in methods not only exclusion criterion, but also the clinical characteristics of patients with CVD to be included (e.g were patients enrolled after how much time from an acute coronary syndrome?)

Response 6: We think this is an excellent suggestion. We revised the detailed inclusion criteria, including (1) age over 18 years old; (2) diagnosed as CVDs, including coronary heart disease (CHD), old myocardial infarction (MI), arrhythmias and heart valve disease; (3) without Percutaneous Coronary Intervention (PCI) or one week after PCI; (4) without Coronary Artery Bypass Graft (CABG) or one month after CABG.

We assessed the clinical characteristics of participants before enrolling in this study. Patients with unstable physical status were not allowed to participate in the experiment, such as unstable angina. If patients have acute coronary syndrome (ACS), they first need to receive medication to a stable state at least 4 weeks and then do exercise rehabilitation. But it still needs the assessment of physical condition and doctor’s opinions. In addition, we updated “Table 1. Characteristics of study participants” since we filled in some clinical information that was missing in our previous draft.

We feel very sorry for our mistake. We really hope to gain your understanding.

Point 7: How was assigned the level of danger?

Response 7: Thank you for the suggestion. The level of danger was based on symptoms and clinical indicators in our study. The following table showed the detailed standards for grading. It was also added to the appendix in our revised manuscript. 

Point 8: Being this a prospective study a sample size calculation should be present in methods.

Response 8: We are very grateful to your professional suggestion. Sample size circulation is necessary for our study. We have added related content in subsection 2.5.

Sample size calculation was based on physical tolerance (primary outcome) measured by oxygen uptake (VO2) at anaerobic threshold (AT). According to previous studies, VO2 at AT increased around 10%. Thus, we set the difference between before and after rehabilitation as 15% in our study. The significance level was 0.05, and the power was 0.90. We used the online free tool from Harvard University to circulate the sample size. We got that 21 was the minimum sample size of each group. But due to self-controlled experiment, the total sample size in our study was at least 21.

The website was “http://hedwig.mgh.harvard.edu/sample_size/js/js_parallel_quant.html”.

Point 9: Please remove this sentence “Authors should discuss the results and how they can be interpreted from the perspective of previous studies and of the working hypotheses. The findings and their implications should be discussed in the broadest context possible. Future research directions may also be highlighted” from the beginning of the discussion.

Response 9: We are really sorry for our careless mistake. Thank you very much for pointing it out. We have removed this sentence at the beginning of discussion.

We sincerely hope that the revised manuscript has addressed your comments and suggestions. Once again, thank you very much for your professional suggestions.

Reviewer 2 Report

Authors Ma et al write a manuscript with their groups experience for cardiac rehabilitation focusing on adaptive posture balance cardiac rehabilitation.  The topic is interesting but the authors need to be clearer in the way this is conveyed.  There are several fragmented sentences, there are abbreviations that should be written out and also at times the writing seems rushed.  I would recommend a very detailed re-review for these issues as well as English grammar and syntax. 

Author Response

Please see the attachment. Thank you for your understanding.

Response to Reviewer 2 Comments

Manuscript ID number: jcm-1866806

Title: Adaptive posture-balance cardiac rehabilitation exercise significantly improved physical tolerance in patients with cardiovascular diseases

Authors: Mei Ma, Bowen Zhang, Xinxin Yan, Xiang Ji, Deyu Qin, Chaodong Pu, Jingxiang Zhao, Qian Zhang, Heinz Lowis, Ting Li *

We are grateful to your professional review work on our article. As you are concerned, there are several problems that need to be addressed. According to your nice suggestions, we have made extensive corrections to our previous draft. The detailed corrections are listed below.

Because almost every sentence was revised in English, it is messy to highlight all revisions. Thus, We highlight all revisions in content by yellow color, and some important revisions in English by blue color. Thanks very much for your understanding.

Point 1: There are several fragmented sentences.

Response 1: We are very sorry for our poor writing. We have carefully checked and revised every sentence in the draft. And we invited our friend who is a native English speaker to polish the manuscript. Because of the many changes, we cannot list them all here. But we have marked them in blue in the revised paper. Thank you for your understanding. We hope the revised manuscript could help you understand the manuscript better.

Some changes are listed here as examples:

  1. Our group in Tianjin Chest Hospital designed a peculiar rehabilitation method based on balance exercise, named as adaptive posture-balance cardiac rehabilitation exercise (APBCRE). It originated from the summary of large number of daily clinical work experience about CVDs and was specifically designed for reducing the risk of falls. Nevertheless, the actual application effect of APBCRE on patients was not definite. Thus, this study aimed to assess whether this self-created therapy is effective on patients with cardiovascular diseases, and observe concrete changes in physiology and behavior.” is changed toBased on balance exercise, a new rehabilitation approach was designed and named as adaptive posture-balance cardiac rehabilitation exercise (APBCRE). The approach was inspired by clinical practice about CVDs, specifically designing to reduce the risk of falls. To better understand the clinical effectiveness of APBCRE on CVDs patients, the current study aimed to assess whether APBCRE could enhance physical tolerance and improve the quality of life in patients with CVDs.in introduction in lines 63-68.
  2. Before being enrolled, all subjects have signed an informed consent form. The whole experiment included one-month APBCRE therapy and two assessments. The one-month APBCRE consisted of twelve exercise sessions, evenly three times per week. At baseline and after rehabilitation, participants underwent one assessment, including cardiopulmonary exercise testing (CPET), resting metabolic rate (RMR) detection, and quality of life questionnaire (QoL).” is changed to “All subjects have signed informed consent forms before being enrolled. The whole experiment included one-month APBCRE and two clinical assessments before and after APBCRE interventions. The one-month APBCRE consisted of twelve exercise sessions, evenly three times per week. Each assessment included cardiopulmonary exercise testing (CPET), resting metabolism rate (RMR) detection, and questionnaires about quality of life (QoL), depression levels and anxiety levels.“ in method in lines 73-78.
  3. Therefore, the different results of two subgroups indicated that this rehabilitation approach had diverse effect modes on different age levels. For lower-age patients, exercise rehabilitation improved musculoskeletal, respiratory function and cardiovascular, while for higher-age patients, the improvement mainly focused on respiratory function and the increment was also relatively limited.” is changed to “The differences between subgroups indicated that APBCRE had various modes of effect for different age levels. For lower-age patients, APBCRE improved both musculoskeletal, respiratory and cardiovascular function. But for higher-age patients, APBCRE mainly enhanced ventilation when sporting rather than directly improving oxygen utilization of skeletal muscle.” in discussion in lines 247-251.

Point 2: There are abbreviations that should be written out and also at times the writing seems rushed.

Response 2: Thank you for the suggestion. We really apologize that some inappropriate abbreviations make you to misunderstand the contents. We tried our best to revise all abbreviations and list all abbreviations below in order of draft.

  • CVDs: Cardiovascular diseases
  • CR: cardiac rehabilitation
  • HIIT: high-intensity interval training
  • HR: heart rate
  • APBCRE: adaptive posture-balance cardiac rehabilitation exercise
  • CPET: cardiopulmonary exercise testing
  • RMR: resting metabolism rate
  • QoL: quality of life
  • VO2: oxygen uptake
  • AT: anaerobic threshold
  • SF-12: 12-item Short Form Survey
  • CHD: coronary heart disease
  • MI: myocardial infarction
  • PCI: Percutaneous Coronary Intervention
  • CABG: Coronary Artery Bypass Graft
  • RER: respiratory exchange ratio
  • VE: ventilation
  • VE/VCO2: VE-VCO2 slope
  • OUES: oxygen uptake efficiency slope
  • VCO2: carbon dioxide output
  • WAT: The power of the bicycle at AT
  • ee: energy expenditure
  • fat: fat energy
  • cho: carbohydrate energy
  • GAD-7: Generalized Anxiety Disorder 7-item
  • PHQ-9: Patient Health Questionnaire-9
  • SD: standard deviation
  • KNN: K-NearestNeighbor
  • BMI: body mass index
  • cpm: counts per minutes
  • VO2peak peak oxygen uptake

We really hope that the revised manuscript has addressed your comments and suggestions. Once again, thank you very much for your professional suggestions.

Reviewer 3 Report

This is an objective and logical contribution to the supportive therapy of heart diseases, as well as respiratory diseases, especially pulmonary hypertension. Targeted and programmed musculature load, improvement of neuroplasticity indirectly help severe heart patients with better load status and life extension. This research confirms the opinion of many doctors that cardiovascular patients should have a physical load program that does not worsen the status of the heart itself (or/and) the lungs, but strengthens the entire system of auxiliary functions of nerves and muscles that relieve some of the heart's functions and reduce resistance to cardiac output. very necessary for the development of a more optimistic attitude in cardiovascular and irreversible respiratory diseases.

Author Response

Thank you for your affirmation of our study. Please see the attachment.

Response to Reviewer 3 Comments

Manuscript ID number: jcm-1866806

Title: Adaptive posture-balance cardiac rehabilitation exercise significantly improved physical tolerance in patients with cardiovascular diseases

Authors: Mei Ma, Bowen Zhang, Xinxin Yan, Xiang Ji, Deyu Qin, Chaodong Pu, Jingxiang Zhao, Qian Zhang, Heinz Lowis, Ting Li *

We are really grateful to your professional review work on our article. And we sincerely appreciate your affirmation and support of our work. In the future, we will design a multi-center randomized controlled trial with more samples to further explore our findings.

Once again, thank you very much for your suggestions.

Round 2

Reviewer 1 Report

The authors replied to all my questions. I have no other comments.